# Evidence Based on an Integrative Analysis of Multi-Omics Data on METTL7A as a Molecular Marker in Pan-Cancer

**DOI:** 10.3390/biom13020195

**Published:** 2023-01-18

**Authors:** Zikai Liu, Yiqun Chen, Tong Shen

**Affiliations:** School of Public Health, Anhui Medical University, Hefei 230032, China

**Keywords:** METTL7A, pan-cancer analysis, biomarker, omics integrative analysis

## Abstract

Methyltransferase-like protein 7A (METTL7A), an RNA N6-methyladenosine (m6A) methyltransferase, has attracted much attention as it has been found to be closely associated with various types of tumorigenesis and progression. This study provides a comprehensive assessment of METTL7A from a pan-cancer perspective using multi-omics data. The gene ontology enrichment analysis of METTL7A-binding proteins revealed a close association with methylation and lipid metabolism. We then explored the expression of METTL7A in normal tissues, cell lines, different subtypes and cancers, and found that METTL7A was differentially expressed in various cancer species, tumor molecular subtypes and immune subtypes. Evaluation of the diagnostic and prognostic value of METTL7A in pan-cancer revealed that METTL7A had high accuracy in tumor prediction. Moreover, the low expression of METTL7A significantly correlated with the poor prognosis, including kidney renal clear cell carcinoma (KIRC), mesothelioma and sarcoma, indicating that METTL7A could be a potential biomarker for tumor diagnosis and prognosis. We focused on KIRC after pre-screening and analyzed its expression and prognostic value in various clinical subgroups. We found that METTL7A was significantly related to tumor stage, metastasis stage, pathologic stage, primary therapy outcome, histologic grade and gender, and that low METTL7A expression was associated with poorer outcomes. Finally, we analyzed the immune infiltration and co-expressed genes of METTL7A as well as the differentially expressed genes in the high and low expression groups. In conclusion, METTL7A is a better molecular marker for pan-cancer diagnosis and prognosis and has high potential as a diagnostic and prognostic biomarker for KIRC.

## 1. Introduction

N6-methyladenosine (m6A) modification has gained a great deal of attention from researchers in recent years [1]. The m6A modifications are the most abundant internal modifications of RNA in eukaryotic cells and play a key role in cancer through a variety of mechanisms [2,3]. The effects of m6A are mediated by methyltransferases (writers), demethylases (erasers) and m6A binding proteins (readers), with the methyltransferase-like (METTL) family playing an important role [4]. The METTL family is a diverse group of methyltransferases consisting of more than 20 members with the ability to alter the conformation, stability and function of genes by catalyzing methylation [5]. Many studies have shown that 17 members of the METTL family can be regarded as oncogenes or tumor suppressors. For example, METTL1 has been shown to act as an oncogene in bladder cancer, hepatocellular carcinoma (HCC), acute myeloid leukemia, and intrahepatic cholangiocarcinoma [6,7,8,9,10]; the METTL3/14 complex acts as a tumor suppressor in thyroid and endometrial cancers [11,12,13]. Nevertheless, our knowledge of METTL family members is still insufficient, with METTL7A being a controversial molecule and associated with multiple cancers. No pan-cancer analysis of METTL7A is currently available.

METTL7A, also known as AAM-B, was originally identified as a lipid droplet–associated protein in Chinese hamster ovary K2 cells by proteomic analysis [14]. As a member of the methyltransferase-like protein family, it is an integral membrane protein anchored into the endoplasmic reticulum membrane and plays a role in lipid metabolism and the generation of functional organelles by recruiting cellular proteins to form lipid droplets [15,16]. METTL7A is thought to be closely associated with the development, migration, drug resistance and prognosis of a variety of tumors, suggesting that it could be used as a potential molecular marker for tumor diagnosis. For example, exposure of adipocytes to multiple myeloma cells enhances METTL7A activity in m6A methylation through zeste homolog 2–mediated protein methylation, which in turn mediates drug resistance in myeloma [17]. METTL7A may also promote methotrexate resistance by activating pro-survival signaling pathways and attenuating the accumulation of reactive oxygen species in choriocarcinoma cells [18]. On the one hand, Guo et al. found that METTL7A expression was downregulated by data analysis in lung cancer, suggesting that METTL7A is associated with the development and progression of lung adenocarcinoma and is a potential candidate target for early diagnosis and treatment of lung adenocarcinoma [19]. On the other hand, the comprehensive bioinformatics showed that METTL7A had good diagnostic values in patients with osteosarcoma and could be a novel biomarker and potential therapeutic target for breast cancer [20,21]. In addition, METTL7A expression was downregulated in papillary thyroid cancer tissues compared with normal thyroid cells, suggesting a possible association with thyroid carcinogenesis [22]. In summary, METTL7A may play an important role in different cancers or different subtypes.

Although the association between METTL7A and cancer is relatively well analyzed, there is not yet a firm understanding of the relationship and mechanisms behind it, and METTL7A may play distinct roles in different cancers due to the heterogeneity and complexity of tumors. To gain a more systematic and comprehensive understanding of the role of METTL7A in cancer, we explored METTL7A expression and its biological functions from a pan-cancer perspective, focused on its diagnostic and prognostic value. The results showed that METTL7A was not only significantly downregulated in 18 types of cancer but also differentially expressed in 7 molecular cancer subtypes and 8 immune cancer subtypes. Further studies showed that METTL7A was highly accurate in the prediction of 12 cancers, and the low expression of METTL7A significantly correlated with the worst overall survival (OS), disease-specific survival (DSS), and progression-free interval (PFI) in kidney renal clear cell carcinoma (KIRC), mesothelioma (MESO) and sarcoma (SARC). After extensive screening, we focused on KIRC to explore the immune infiltration and co-expression genes of METTL7A in this cancer, as well as the function of the differentially expressed genes (DEGs) in the high-expression and low-expression groups of METTL7A. Based on the above evidence, the biomarker of METTL7A as a diagnostic and prognostic indicator in pan-cancer deserves our attention, and it is also a molecular target worth exploring for KIRC.

## 2. Materials and Methods

### 2.1. Protein–Protein Interaction Network and Enrichment Analysis of METTL7A-Binding Proteins

A total of 40 METTL7A-binding proteins were acquired from the STRING web (https://cn.string-db.org/, accessed on 17 October 2022.) by setting the following main parameters: action–interaction sources (“text mining, experiments, database”), minimum required interaction score (“medium confidence (0.400)”) and maximum number of interactors to show “no more than 50 interactors.” Next, we used Cytoscape (version 3.9.1) to visualize the protein–protein interaction (PPI) network. The Gene Ontology enrichment analysis was conducted for 40 METTL7A-binding proteins using the cluster Profiler package (version 3.14.3) for enrichment analysis and the ggplot2 (version 3.3.3) package for visualization [23].

### 2.2. METTL7A Expression Analysis

We downloaded RNA-seq data from The Cancer Genome Atlas (TCGA) database and the Genotype-Tissue Expression (GTEx) database by UCSC XENA (https://xenabrowser.net/datapages/, accessed on 1 October 2022.), including 7568 normal samples from the GTEx database, 727 para-carcinoma samples and 9807 tumor samples from the TCGA database. RNA-seq data in TPM format were processed uniformly by the Toil process [24,25]. Cell line information was downloaded from the Human Protein Atlas (HPA) database (https://www.proteinatlas.org, accessed on 22 October 2022.). Statistical analysis was performed using R software v3.6.3 and visualization was performed using the ggplot2 package. Wilcoxon rank sum test was used to detect both data sets, and *p* < 0.05 was considered statistically significant (ns, *p* ≥ 0.05; *, *p* < 0.05; **, *p* < 0.01; ***, *p* < 0.001).

### 2.3. METTL7A Expression in Subtypes of Cancers

Correlations between METTL7A expression and molecular subtypes and immune subtypes in pan-cancer were explored based on the TISIDB database [26], which describes tumor-immune interactions by integrating research articles and multiple types of high-throughput data. We also explored the correlation between METTL7A expression and immunomodulators in pan-cancer from the TISIDB database.

### 2.4. Diagnostic Value and Prognostic Value Analysis in Pan-Cancer

The diagnostic value of METTL7A in pan-cancer was evaluated using the receiver operating characteristic (ROC) curve. The closer the area under the curve (AUC) is to 1, the better the diagnosis. There is a certain accuracy when the AUC is between 0.7 and 0.9, with a high accuracy when it is greater than 0.9. The relationship between METTL7A expression and cancer prognosis (OS, DSS and PFI) was assessed using Kaplan–Meier plots. The prognostic data are from a previous study [27]. The survminer package was used for visualization and the survival package for statistical analysis. The Cox regression was used for hypothesis testing, and differences were considered statistically significant at *p* < 0.05.

### 2.5. METTL7A Expression and Prognostic Value Analysis in Various Clinical Subgroups of KIRC

The box plots and tables were presented for METTL7A expression levels of patients with various clinical characteristics in UCEC. The RNA-seq data and related clinical data in level 3 HTSeq-fragments per kilobase per million (FPKM) formats were downloaded from the TCGA database, converted to transcripts per million reads (TPM) format, and then analyzed after log2 conversion. We used the Wilcoxon rank sum test to detect two sets of data, and *p* < 0.05 was considered statistically significant (ns, *p* ≥ 0.05; *, *p* < 0.05; **, *p* < 0.01; ***, *p* < 0.001). In addition, we further investigated the associations between METTL7A expression and prognosis (OS, DSS and PFI) in various clinical subgroups of KIRC.

### 2.6. Immune Infiltration and Co-Expression Gene Analysis of METTL7A in KIRC

Immune infiltration analysis of METTL7A was conducted by single-sample gene set enrichment analysis (SSGSEA) using GSVA package in R (3.6.3). A total of 24 types of infiltrating immune cells were obtained from previous studies [28]. Spearman correction was used to analyze the correlation between METTL7A and the enrichment scores of 24 types of immune cells. The top 50 co-expressed genes in KIRC that were positively and negatively correlated with METTL7A expression were obtained. The stat package was used to display the gene co-expression heatmap. The Pearson correlation coefficient was used to show the correlations of the top 10 gene expressions in the heatmap with METTL7A.

### 2.7. DEGs between METTL7A High Expression and Low Expression Groups in KIRC

We explored the DEGs between different METTL7A expression groups (low expression group: 0–50%; high expression group: 50–100%) in KIRC using the DESeq2 (version 1.26.0) package [29]. The ggplot2 package was used to draw the volcano map (|log2 fold-change (FC)| > 1.5, *p* < 0.05). Then we used the cluster profiler package to execute GO and KEGG functional enrichment analysis, and the ggplot2 package was used for visualization. Furthermore, we built a PPI network of DEGs using STRING web and analyzed the hub gene by the MCC algorithm of CytoHubba in Cytoscape (version 3.9.1).

## 3. Results

### 3.1. PPI Network and GO Enrichment Analysis of METTL7A-Binding Proteins

The STRING database and Cytoscape were used to find 40 targeted binding proteins of METTL7A (Figure 1A) (Appendix A). Subsequently, we conducted the GO enrichment analysis of 40 targeted binding proteins, revealing that the primary biological process (BP) comprised methylation, lipid droplet organization, retrograde protein transport, ER to cytosol, and S-adenosylmethionine metabolic process. The cellular component (CC) was primarily involved in lipid droplets. The molecular function (MF) was mainly enriched in transferase activity, transferring one-carbon groups, and methyltransferase activity (Figure 1B,C).

### 3.2. METTL7A Expression Analysis

We exhibited METTL7A expression in normal tissues from the GTEx database and found that METTL7A was expressed in all tissues, with the five most highly expressed tissues being the liver, thyroid gland, adipose tissue, stomach and adrenal gland (Figure 2A). We also detected METTL7A expression in many cell lines, and the highest expression cell line was mesenchymal (Figure 2B). For TCGA tumors and adjacent normal tissues, METTL7A expression was significantly downregulated in 15 cancer types, including breast invasive carcinoma (BRCA), cholangiocarcinoma (CHOL), colon adenocarcinoma (COAD), esophageal carcinoma (ESCA), head and neck squamous cell carcinoma (HNSC), kidney chromophobe (KICH), KIRC, kidney renal papillary cell carcinoma (KIRP), lung adenocarcinoma (LUAD), lung squamous cell carcinoma (LUSC), prostate adenocarcinoma (PRAD), rectum adenocarcinoma (READ), stomach adenocarcinoma (STAD), thyroid carcinoma (THCA), and uterine corpus endometrial carcinoma (UCEC) (Figure 2C). In the meantime, the data of tumors and normal tissues from the TCGA and GTEx database showed that METTL7A expression was significantly downregulated in 18 cancer types, including bladder urothelial carcinoma (BLCA), BRCA, cervical squamous cell carcinoma and endocervical adenocarcinoma (CESC), CHOL, COAD, ESCA, HNSC, KIRC, LUAD, LUSC, ovarian serous cystadenocarcinoma (OV), PRAD, READ, skin cutaneous melanoma (SKCM), STAD, THCA, UCEC, and uterine carcinosarcoma (UCS), while it was upregulated in 8 cancer types, including adrenocortical carcinoma (ACC), lymphoid neoplasm diffuse large B-cell lymphoma (DLBC), glioblastoma multiforme (GBM), acute myeloid leukemia (LAML), brain lower grade glioma (LGG), liver hepatocellular carcinoma (LIHC), pancreatic adenocarcinoma (PAAD), and thymoma (THYM) (Figure 2D).

### 3.3. METTL7A Expression in Molecular or Immune Subtype of Cancers

We explored the correlations between METTL7A differential expression and pan-cancer molecular subtypes from the TISIDB database and found that METTL7A was expressed differently in molecular subtypes of 7 cancer types, including COAD, ESCA, HNSC, LGG, pheochromocytoma and paraganglioma (PCPG), READ, and STAD. Moreover, for COAD, the highest expression level of METTL7A among the molecular subtypes was genomically stable (GS) (Figure 3A). For ESCA, the highest expression level of METTL7A among the molecular subtypes was chromosomal instability (CIN) (Figure 3B). For HNSC, the highest expression level of METTL7A among the molecular subtypes was atypical (Figure 3C). For LGG, the highest expression level of METTL7A among the molecular subtypes was mesenchymal-like (Figure 3D). For PCPG, the highest expression level of METTL7A among the molecular subtypes was pseudohypoxia (Figure 3E). For READ, the highest expression level of METTL7A among the molecular subtypes was GS (Figure 3F). For STAD, the highest expression level of METTL7A among the molecular subtypes was GS (Figure 3G).

Meanwhile, we observed that METTL7A expression was significantly associated with different immune subtypes of 8 cancer types, including BRCA (Figure 3H), CHOL (Figure 3I), KIRC (Figure 3J), LUAD (Figure 3K), MESO (Figure 3L), OV (Figure 3M), PAAD (Figure 3N), and SARC (Figure 3O). At the same time, we observed that the expression of METTL7A in malignant tumors was related to immune stimulators (Appendix A) and immune inhibitors (Appendix A).

### 3.4. Diagnostic Value of METTL7A in Pan-Cancer

The ROC curve was used to assess the diagnostic value of METTL7A in pan-cancer. The results showed that METTL7A had a certain accuracy (AUC > 0.7) in predicting 21 cancer types. Among them, METTL7A had a high accuracy (AUC > 0.9) in predicting 12 cancer types, including BRCA (AUC = 0.944, CI: 0.931–0.957) (Figure 4A), COAD (AUC = 0.966, CI: 0.950–0.981) (Figure 4B), DLBC (AUC = 0.968, CI: 0.950–0.985) (Figure 4C), LAML (AUC = 0.950, CI: 0.922–0.979) (Figure 4D), LGG (AUC = 0.941, CI: 0.928–0.955) (Figure 4E), LUAD (AUC = 0.919, CI: 0.900–0.938) (Figure 4F), LUSC (AUC = 0.928, CI: 0.910–0.946) (Figure 4G), OV (AUC = 0.957, CI: 0.939–0.975) (Figure 4H), READ (AUC = 0.958, CI: 0.934–0.983) (Figure 4I), STAD (AUC = 0.919, CI: 0.896–0.942) (Figure 4J), UCEC (AUC = 0.957, CI: 0.935–0.976) (Figure 4K), and UCS (AUC = 0.967, CI: 0.944–0.991) (Figure 4L).

### 3.5. Prognostic Value of METTL7A in Pan-Cancer

METTL7A expression was significantly correlated with OS, DSS, and PFI of KIRC, MESO, and SARC. For KIRC, Cox regression results showed that a worse prognosis was correlated with lower METTL7A expression, including OS (hazard ratio (HR) = 0.38, 95% confidence interval (CI): 0.27–0.53, *p* < 0.001) (Figure 5A), DSS (HR = 0.26, 95% CI: 0.17–0.42, *p* < 0.001) (Figure 5B), and PFI (HR = 0.41, 95% CI: 0.30–0.58, *p* < 0.001) (Figure 5C). For MESO and SARC, the same conclusion could be obtained that the downregulation of METTL7A predicted poor prognosis. For MESO, OS (HR = 0.34, 95% CI: 0.21–0.56, *p* < 0.001) (Figure 5D), DSS (HR = 0.26, 95% CI: 0.13–0.48, *p* < 0.001) (Figure 5E), and PFI (HR = 0.27, 95% CI: 0.15–0.47, *p* < 0.001) (Figure 5F). For SARC, OS (HR = 0.51, 95% CI: 0.34–0.77, *p* = 0.001) (Figure 5G), DSS (HR = 0.51, 95% CI: 0.33–0.79, *p* = 0.003) (Figure 5H), and PFI (HR = 0.64, 95% CI: 0.46–0.89, *p* = 0.009) (Figure 5I).

### 3.6. METTL7A Expression and Prognostic Value Analysis in Various Clinical Subgroups of KIRC

We investigated the relationship between METTL7A and various clinical subgroups in KIRC and found that the expression of METTL7A was significantly related to tumor stage (T stage), metastasis stage (M stage), pathologic stage, primary therapy outcome, histologic grade, and gender of KIRC (Table 1). Among them, with the intensification of disease severity, the expression of METTL7A in patients gradually decreased, such as T4 (Figure 6A), M1 (Figure 6B), and stage III/IV (Figure 6C). Similarly, METTL7A was expressed lower in primary therapy outcome (PD) (Figure 6D) and histologic grade 4 (Figure 6E).

Further, we explored the correlations between METTL7A and prognosis (OS, DSS, PFI) in various clinical subgroups of KIRC. The results showed that the lower expression of METTL7A had a worse OS in most clinical subgroups, including the subgroup of T3 stage (Figure 6G), the subgroup of M1 stage (Figure 6H), the subgroup of pathologic stage IV (Figure 6I), the subgroup of histologic grade (G3) (Figure 6J), and the subgroup of age > 60 (Figure 6K). For DSS, the lower expression of METTL7A had a worse DSS in the subgroup of T3 stage (Figure 6L), the subgroup of M1 stage (Figure 6M), the subgroup of pathologic stage IV (Figure 6N), the subgroup of histologic grade (G3) (Figure 6O), and the subgroup of age > 60 (Figure 6P). For PFI, the lower expression of METTL7A had a worse PFI in the subgroup of histologic grade (G3) (Figure 6Q) and the subgroup of age > 60 (Figure 6R).

### 3.7. Immune Infiltration and Co-Expression Gene Analysis of METTL7A in KIRC

The expression level of METTL7A in the KIRC microenvironment was correlated with the immune cell infiltration level quantified by SSGSEA. Notably, METTL7A was negatively associated with Treg and NK CD56bright cells (Figure 7A).

Additionally, we screened the top 50 co-expression genes that were positively or negatively correlated with METTL7A expression in KICE, presented in heatmap. In addition, METTL7A expression and the top 10 gene expressions were also explored. For positive correlation (Figure 7B), the top 10 genes were RBM47 (r = 0.751), TMEM192 (r = 0.730), SOWAHB (r = 0.723), CAT (r = 0.721), C11orf54 (r = 0.707), PANK1 (r = 0.707), ETFDH (r = 0.701), ASB8 (r = 0.701), ABHD6 (r = 0.697), EHHADH (r = 0.696) (Appendix A). For negative correlation (Figure 7C), the top 10 genes were C20orf141 (r = −0.458), TUBB3 (r = −0.444), CENPW (r = −0.429), PPP1R14B (r = −0.428), C4orf48 (r = −0.424), TMEM158 (r = −0.424), SH3BGRL3 (r = −0.423), NIPAL4 (r = −0.423), SPOCD1 (r = −0.423), and UBE2C (r = −0.421) (Appendix A).

### 3.8. DEGs between METTL7A High and Low Expression Groups in KIRC

A total of 311 DEGs were acquired with the threshold values of |log2 fold change (FC)| > 1.5 and adjusted *p*-value < 0.05, including 50 upregulated genes and 261 downregulated genes (Figure 8A). Subsequently, we performed the GO and KEGG enrichment analysis for upregulated and downregulated DEGs, respectively (Figure 8B,C). For upregulated DEGs, the BP was mainly enriched in anion transmembrane transport, hormone transport and chloride transmembrane transport. The CC was primarily involved in basolateral plasma membrane, apical part of cell and apical plasma membrane. The MF was mainly enriched in anion transmembrane transporter activity, passive transmembrane transporter activity and ion channel activity. The KEGG pathway enriched was mainly related to bile secretion, gastric acid secretion and collecting duct acid secretion (Figure 8D). For downregulated DEGs, the BP was mainly enriched in epidermis development, negative regulation of hydrolase activity and proteolysis. The CC was primarily involved in protein–DNA complex, DNA packaging complex and nucleosome. The MF was mainly enriched in receptor ligand activity, enzyme inhibitor activity and G protein–coupled receptor binding. The KEGG pathway enriched was mainly related to transcriptional misregulation in cancer, estrogen signaling pathway and IL-17 signaling pathway (Figure 8E). Finally, we obtained the top 8 hub genes of 311 DEGs (Appendix A), including HIST1H2BM, HIST1H1B, HIST1H4F, HIST1H2BC, HIST1H1E, HIST1H3J, HIST2H2AB, and HISTAH2AI. Among them, the top three hub genes were HIST1H2BM, HIST1H1B, and HIST1H4F.

## 4. Discussion

The METTL family is a diverse group of methyltransferases with the ability to methylate nucleotides, proteins and small-molecule metabolites, and it has gained interest in recent years due to its association with tumors [30]. As a member of the METTL family, METTL7A is an m6A methyltransferase whose biological role is to recruit cellular proteins to form lipid droplets [17,31]. METTL7A is thought to be associated with a variety of cancers, but its role in different cancer species is not the same. It has been shown to be an oncogene in multiple myeloma and choriocarcinoma cells [17,18] but as a protective factor in breast cancer cells and liver cancer [32,33]. Much evidence suggests that METTL7A is a molecular marker that is well worth exploring in the pan-cancer field.

In view of the absence of systematic pan-cancer analysis of METTL7A, a comprehensive evaluation of METTL7A was conducted using multi-omics data. First, for a better understanding of the biofunction of METTL7A, we obtained 40 targeted binding proteins of METTL7A using the STRING database, and their GO enrichment analysis showed that METTL7A and its targeted binding proteins were associated with methylation and lipid metabolism. Subsequently, METTL7A expression was evaluated in normal, tumor and paraneoplastic tissues by using the HPA database, TCGA database, and GTEx database, and the results showed that the top five tissues with the highest METTL7A expression in normal tissues were the liver, thyroid gland, adipose tissue, stomach and adrenal gland. In addition, as seen in the comparison of tumor and normal tissues, the expression level of METTL7A was significantly downregulated in 18 cancer types. METTL7A is often regarded as a cancer suppressor, but there is not yet a clear mechanism to explain it. In an experiment in HCC cell lines, overexpression of METTL7A significantly reduced the cell viability, the frequency of foci formation and the number of colonies formed in soft agar compared with other groups, and, conversely, METTL7A silencing enhanced cell tumorigenicity [33]. On the other hand, it has been suggested that the tumor-suppressive effect exerted by METTL7A may be related to the Golgi apparatus. Many diseases, including cancer, are associated with the loss and fragmentation of the Golgi ribbon [34]. Silencing of METTL7A can cause Golgi rupture and further induce tumor development, whereas METTL7A overexpression can reverse this effect by interacting with METTL7B [35]. Although the mechanism of fragmentation of the Golgi apparatus caused by METTL7A is not clear, it may explain the decrease of METTL7A expression in cancer tissues to a certain extent. Interestingly, we also found that METTL7A expression was significantly upregulated in eight cancer species, which may be related to its mediated promotion of drug resistance [17,18], but further experimental design is needed to explore the exact mechanism.

Furthermore, we explored the correlations between METTL7A differential expression and pan-cancer molecular subtypes from the TISIDB database and found that METTL7A was expressed differently in molecular subtypes of 7 cancer types, while METTL7A expression was significantly associated with different immune subtypes of 8 cancer types, and the expression of METTL7A in KIRC was significantly higher in immunologically quiet subgroups than in other immune subgroups, further confirming the association between METTL7A and different cancer species. We then immediately evaluated the diagnostic and prognostic value of METTL7A in pan-cancer. The ROC curve showed that METTL7A had a certain accuracy (AUC > 0.7) in predicting 21 cancer types, including KIRC (AUC = 0.813) (Appendix A). More exaggeratedly, METTL7A had a high accuracy (AUC > 0.9) in predicting 12 cancer types, including BRCA, COAD, DLBC, LAML, LGG, LUAD, LUSC, OV, READ, STAD, UCEC, and UCS, suggesting that METTL7A may be a very promising molecular indicator for diagnosis. When evaluating the prognostic value of METTL7A expression level, we were surprised to find that the low expression of METTL7A significantly correlated with the worst OS, DSS, and PFI of KIRC, MESO, and SARC, suggesting that METTL7A is also an excellent prognostic indicator.

After preliminary screening, we decided to take KIRC as the main research object for the next analysis. KIRC is a common genitourinary malignancy, accounting for 80% of renal malignancies [36,37]. Although the overall 5-year survival rate is approximately 75%, the treatment of advanced and metastatic KIRC remains a worldwide challenge [38]. Due to limited treatment options, the search for new diagnostic and prognostic markers of KIRC is interesting. We analyzed it from the perspective of expression and prognostic value of different clinical subgroups and found that METTL7A was significantly related to tumor stage (T stage), metastasis stage (M stage), pathologic stage, primary therapy outcome, histologic grade and gender of KIRC. Interestingly, METTL7A expression gradually decreased with increasing disease progression, showing a negative correlation and suggesting that low METTL7A expression was associated with poorer prognosis. Furthermore, we performed immune infiltration analysis as well as co-expression gene analysis to complement the evidence for the association of METTL7A expression with KIRC. The results of immune infiltration analysis showed that METTL7A expression in the KIRC microenvironment was negatively correlated with Treg and NK CD56bright cells. Furthermore, the results of co-expression gene analysis showed that the positive correlated top 10 co-expression genes with METTL7A in KIRC were RBM47, TMEM192, SOWAHB, CAT, C11orf54, PANK1, ETFDH, ASB8, ABHD6, and EHHADH. The negative correlated top 10 co-expression genes with METTL7A were C20orf141, TUBB3, CENPW, PPP1R14B, C4orf48, TMEM158, SH3BGRL3, NIPAL4, SPOCD1, and UBE2C. Finally, we analyzed the DEGs between METTL7A high- and low-expression groups in KIRC, then performed GO functional enrichment and KEGG pathway analysis of the DEGs. This suggests that DEGs are associated with nucleic acid transcription, transmembrane transport, and epidermal function, and they may be regulated by the IL-17 signaling pathway. Finally, we identified the hub genes of DEGs, including HIST1H2BM, HIST1H1B, HIST1H4F, HIST1H2BC, and HIST1H1E.

Based on multiple databases, including the TISIDB database, TCGA database, GTEx database and STRING database, this study utilized a variety of research methods to evaluate the expression of METTL7A in tumors and was more comprehensive than previous studies on METTL7A analysis. We included a number of tumors in the study, systematically evaluated the diagnostic and prognostic value of METTL7A in cancer, and focused on KIRC in depth. This work extensively assessed the pan-cancer value of METTL7A and made a systematic evaluation of the relationship between METTL7A and KIRC, filling a gap in this research field. Nevertheless, the absence of experimental validation is a limitation of this study. We will make this a focus of our future work and carry out relevant work for experimental validation in the following period. In addition, considering the endless emergence of new bioinformatics analysis techniques [39,40], we will consider further studies of the significance of METTL7A using the machine learning methods combined with Mendelian randomization.

In conclusion, METTL7A is a better molecular marker with diagnostic and prognostic value in pan-cancer, especially in KIRC. Low expression of METTL7A is significantly associated with poor prognosis of KIRC and is a predictive molecule with a high degree of accuracy. This study provides a basis for a more comprehensive analysis of the future clinical application of this molecule in oncology therapy.

## Figures and Tables

**Figure 1 biomolecules-13-00195-f001:**
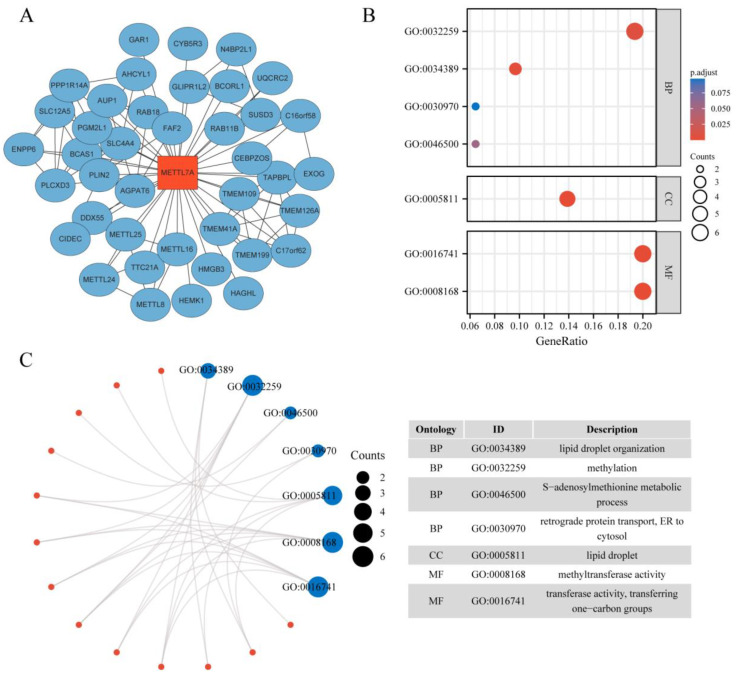
Protein–protein interaction (PPI) network and GO analysis of 40 targeted binding proteins of METTL7A. (**A**) PPI network; (**B**) GO analysis; (**C**) visual network of GO analysis (red: molecular; blue: enrichment results).

**Figure 2 biomolecules-13-00195-f002:**
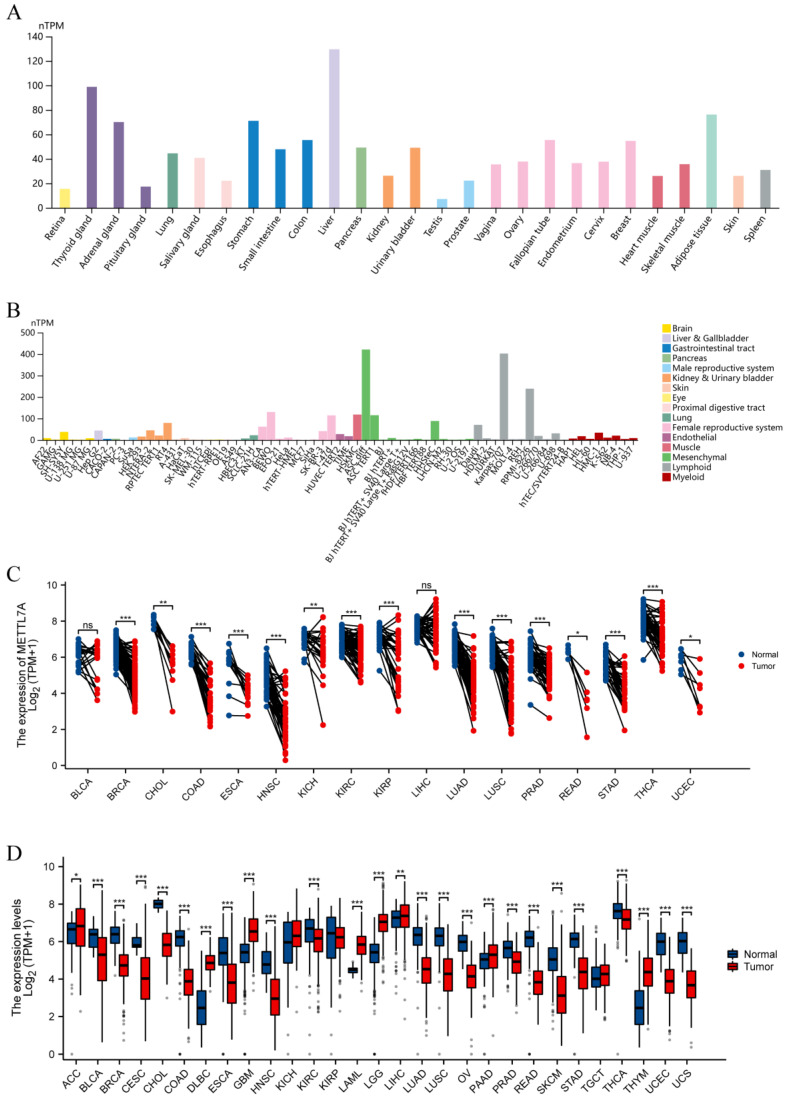
Expression level of METTL7A gene in tumor and normal tissues. (**A**) METTL7A expression in normal tissues; (**B**) METTL7A expression in cell lines; (**C**) METTL7A expression in TCGA tumors and adjacent normal tissues; (**D**) METTL7A expression in TCGA tumors and normal tissues with the data of the GTEx database as controls (grey dot: outlier) (ns, *p* ≥ 0.05; *, *p* < 0.05; **, *p* < 0.01; ***, *p* < 0.001).

**Figure 3 biomolecules-13-00195-f003:**
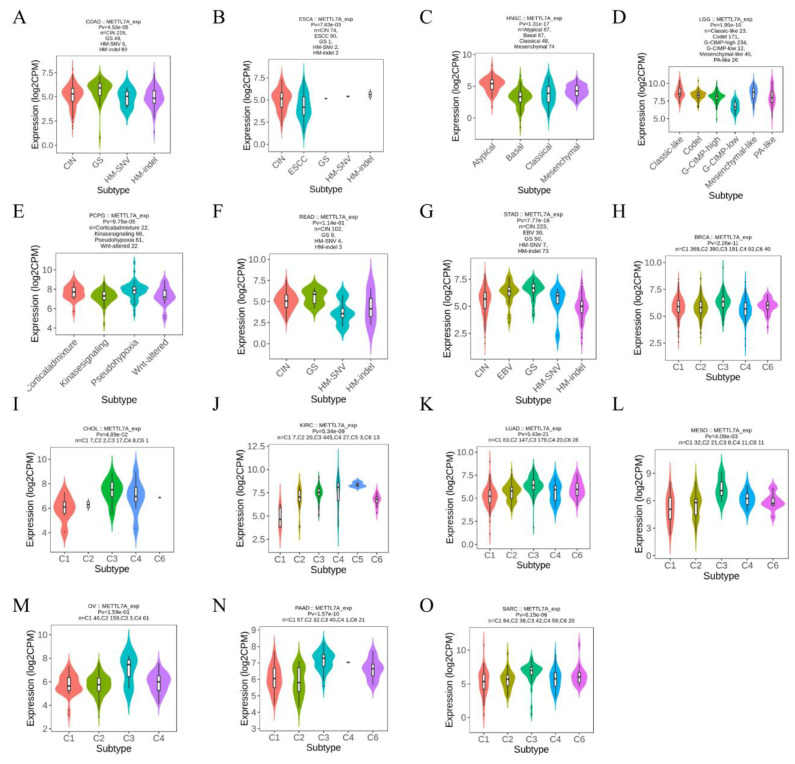
Correlation between METTL7A expression and molecular or immune subtypes across TCGA tumors. (**A**–**G**) molecular subtypes. (**A**) colon adenocarcinoma (COAD); (**B**) esophageal carcinoma (ESCA); (**C**) head and neck squamous cell carcinoma (HNSC); (**D**) brain lower grade glioma (LGG); (**E**) pheochromocytoma and paraganglioma (PCPG); (**F**) rectum adenocarcinoma (READ); (**G**) stomach adenocarcinoma (STAD). (**H**–**O**) immune subtypes. (**H**) breast invasive carcinoma (BRCA); (**I**) cholangiocarcinoma (CHOL); (**J**) kidney renal clear cell carcinoma (KIRC); (**K**) lung adenocarcinoma (LUAD); (**L**) mesothelioma (MESO); (**M**) ovarian serous cystadenocarcinoma (OV); (**N**) pancreatic adenocarcinoma (PAAD); (**O**) sarcoma (SARC). C1: wound healing, C2: IFN-gamma dominant, C3: inflammatory, C4: lymphocyte depleted, C5: immunologically quiet, C6: TGF-b dominant.

**Figure 4 biomolecules-13-00195-f004:**
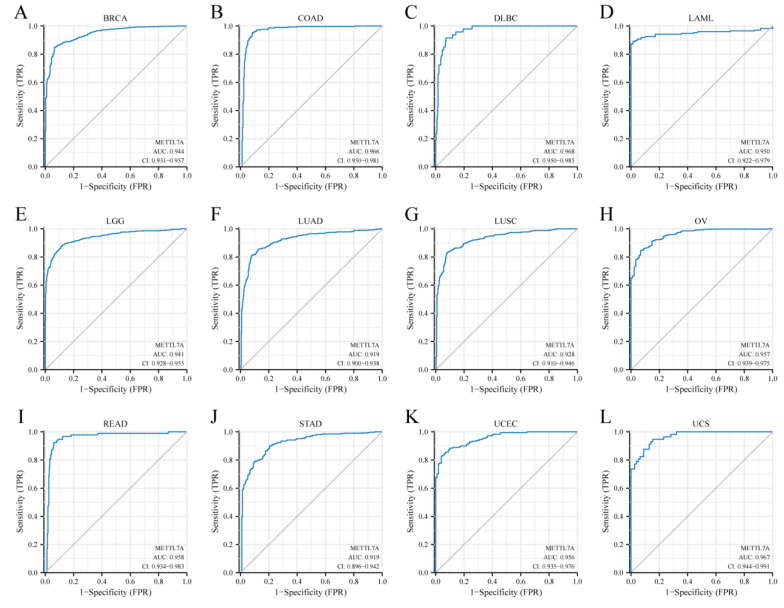
Receiver operating characteristic (ROC) curve for METTL7A expression in pan-cancer. (**A**) breast invasive carcinoma (BRCA); (**B**) colon adenocarcinoma (COAD); (**C**) lymphoid neoplasm diffuse large B-cell lymphoma (DLBC); (**D**) acute myeloid leukemia (LAML); (**E**) brain lower grade glioma (LGG); (**F**) lung adenocarcinoma (LUAD); (**G**) lung squamous cell carcinoma (LUSC); (**H**) ovarian serous cystadenocarcinoma (OV); (**I**) rectum adenocarcinoma (READ); (**J**) stomach adenocarcinoma (STAD); (**K**) uterine corpus endometrial carcinoma (UCEC); (**L**) uterine carcinosarcoma (UCS).

**Figure 5 biomolecules-13-00195-f005:**
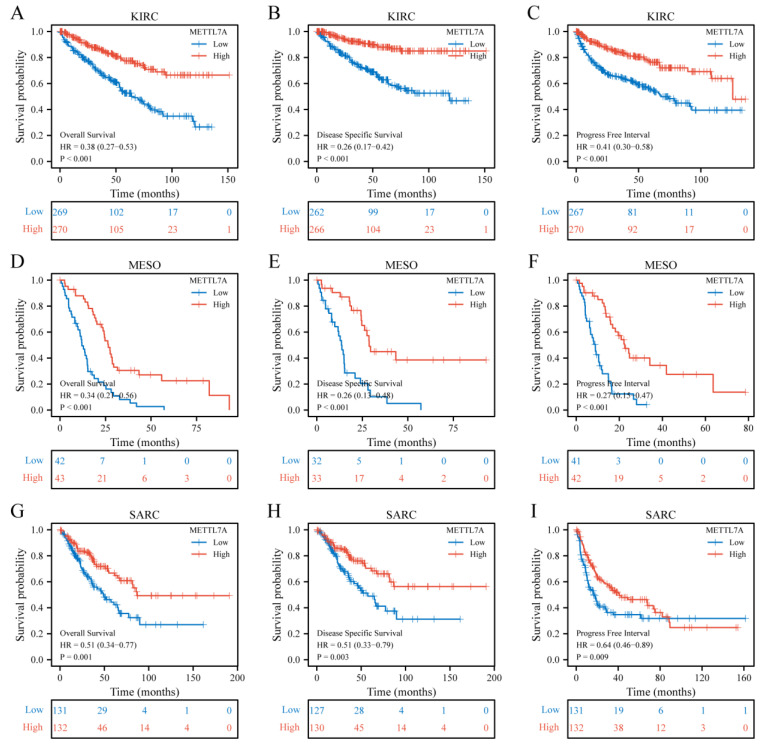
Correlations between METTL7A expression and the prognosis (OS, DSS, and PFI) of cancers. (**A**–**C**) kidney renal clear cell carcinoma (KIRC); (**D**–**F**) mesothelioma (MESO); (**G**–**I**) sarcoma (SARC).

**Figure 6 biomolecules-13-00195-f006:**
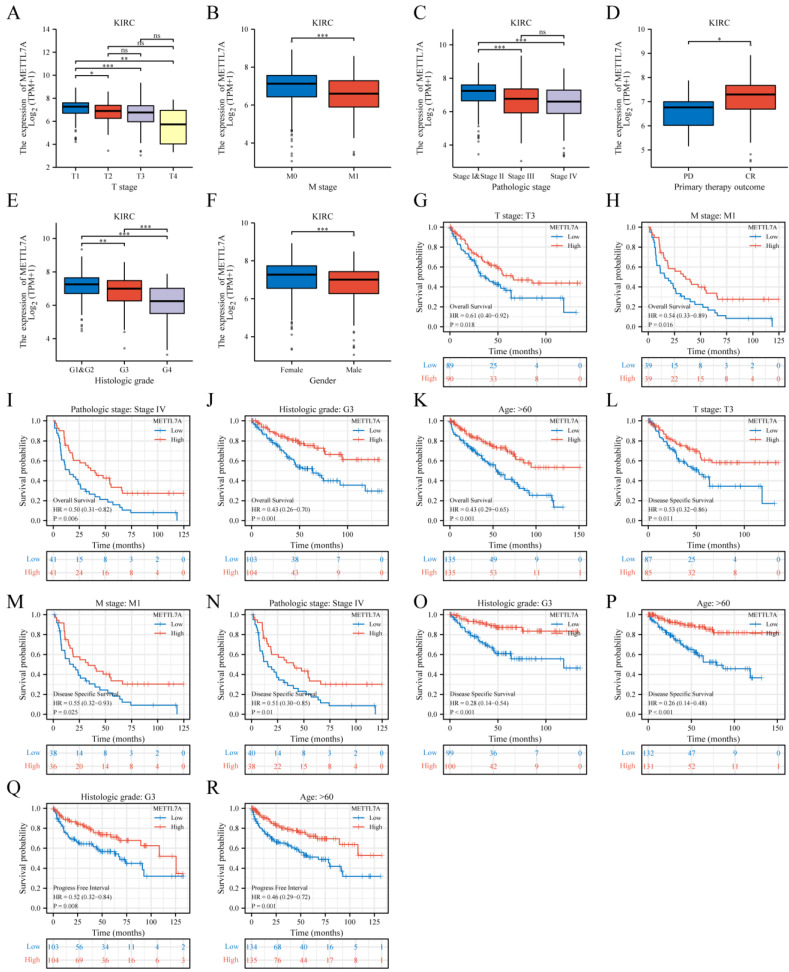
METTL7A expression and prognostic value analysis in various clinical subgroups of KIRC. (**A**–**F**): METTL7A expression in different clinical subgroups. (**A**) T stage; (**B**) M stage; (**C**) pathologic stage; (**D**) primary therapy outcome; (**E**) histologic grade; (**F**) gender (grey dot: outlier) (ns, *p* ≥ 0.05; *, *p* < 0.05; **, *p* < 0.01; ***, *p* < 0.001). (**G**–**R**) prognostic value analysis in different clinical subgroups. (**G**–**K**) OS; (**L**–**P**) DSS; (**Q**–**R**) PFI.

**Figure 7 biomolecules-13-00195-f007:**
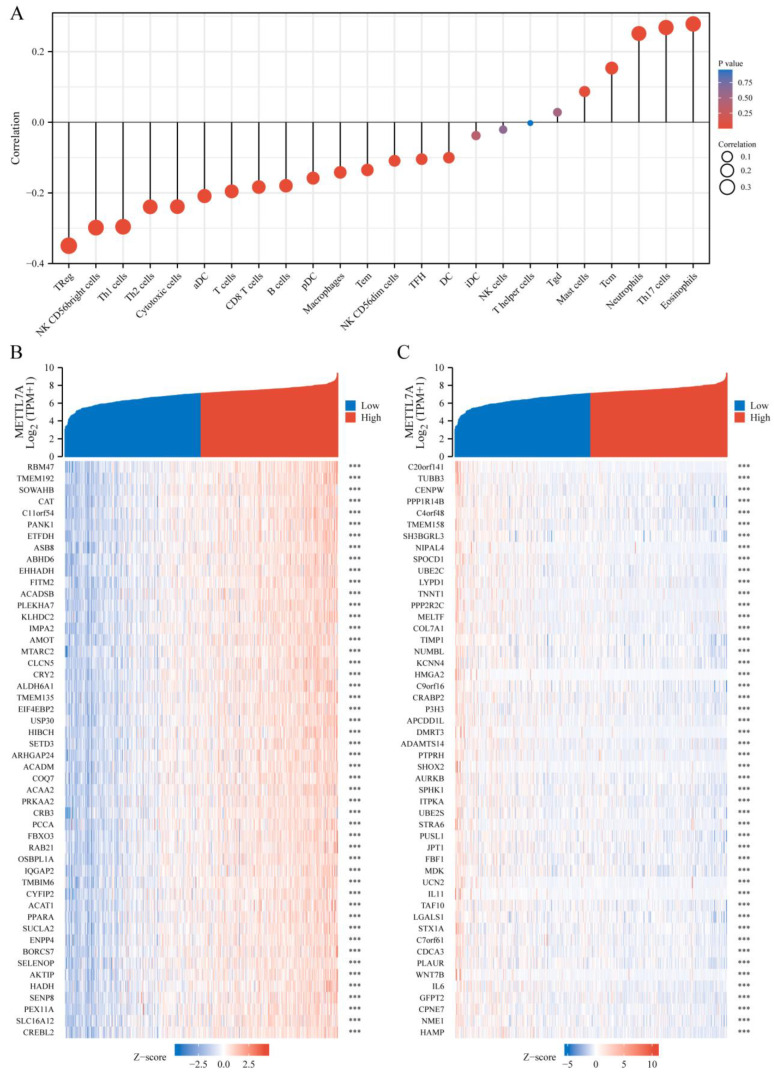
Immune infiltration and co-expression gene analysis of METTL7A in KIRC. (**A**) the forest plots show the correlation between METTL7A and immune cells. (**B**) the gene co-expression heatmap of the positive correlation with METTL7A in KIRC. (**C**) the gene co-expression heatmap of the negative correlation with METTL7A in KIRC (***, *p* < 0.001).

**Figure 8 biomolecules-13-00195-f008:**
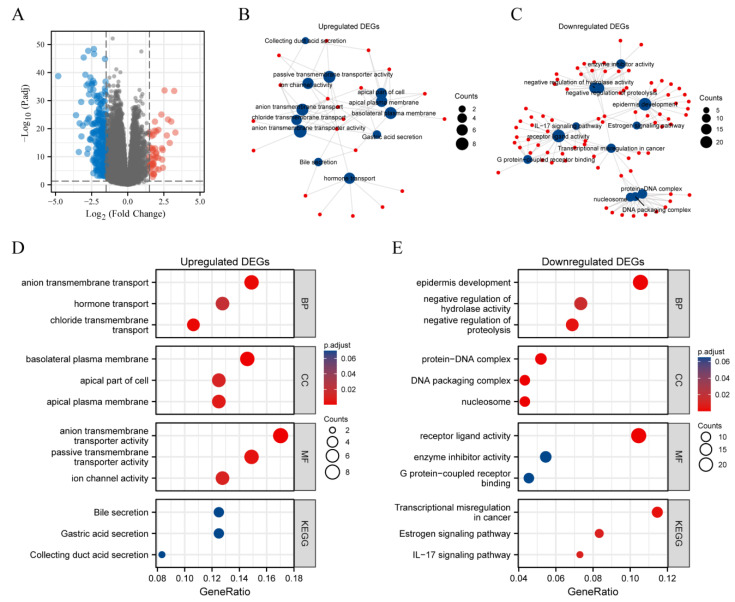
The volcano map and enrichment analysis of DEGs between METTL7A high- and low-expression groups in KIRC. (**A**) The volcano map of DEGs (red: upregulation; blue: downregulation); (**B**,**C**) Visual networks for enrichment analysis (red: molecular; blue: enrichment results); (**D**) GO and KEGG analysis of upregulated DEGs; (**E**) GO and KEGG analysis of downregulated DEGs.

**Table 1 biomolecules-13-00195-t001:** Clinical characteristics of KIRC patients.

Characteristic	Low Expression of METTL7A	High Expression of METTL7A	*p*
*n*	269	270	
T stage	*n* (%)		<0.001
T1	108 (20%)	170 (31.5%)	
T2	38 (7.1%)	33 (6.1%)	
T3	115 (21.3%)	64 (11.9%)	
T4	8 (1.5%)	3 (0.6%)	
M stage	*n* (%)		<0.001
M0	202 (39.9%)	226 (44.7%)	
M1	54 (10.7%)	24 (4.7%)	
Pathologic stage	*n* (%)		<0.001
Stage I	105 (19.6%)	167 (31.2%)	
Stage II	29 (5.4%)	30 (5.6%)	
Stage III	77 (14.4%)	46 (8.6%)	
Stage IV	56 (10.4%)	26 (4.9%)	
Primary therapy outcome	*n* (%)		0.177
PD	8 (5.4%)	3 (2%)	
CR	52 (35.4%)	76 (51.7%)	
Histologic grade	*n* (%)		<0.001
G1	3 (0.6%)	11 (2.1%)	
G2	97 (18.3%)	138 (26%)	
G3	107 (20.2%)	100 (18.8%)	
G4	60 (11.3%)	15 (2.8%)	
Gender	*n* (%)		0.025
Female	80 (14.8%)	106 (19.7%)	
Male	189 (35.1%)	164 (30.4%)	

## Data Availability

The data presented in this study are available in the main text, figures, tables and Appendix A.

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
