# Peer review of "Evidence Based on an Integrative Analysis of Multi-Omics Data on METTL7A as a Molecular Marker in Pan-Cancer"

_biomolecules, 2023, doi:10.3390/biom13020195_

Round 1

Reviewer 1 Report

Zikai Liu et al reported new finding that METTL7A is a molecular marker in pan-cancer by integrative analysis. The METTL7A is an RNA N6-methyladenosine (m6A) methyltransferase, which is associated with tumorigenesis. The authors performed comprehensive assessment of METTL7A by multi-omics data. Their results showed that the METTL7A has a close association with methylation and lipid metabolism, and the METTL7A is indicator of tumor prediction and correlated with the prognosis of kidney renal clear cell carcinoma, mesothelioma and sarcoma. The expression level is related with tumor stage, metastasis stage, primary therapy outcome, histologic grade etc. They reported that low expression level of METTL7A is associated with poorer outcomes. The authors concluded that METTL7A is important molecular marker of diagnostic and prognostic and has potential biomarker of KIRC. It is interesting study and important finding for biomarker of tumor.

There are some concerns as follow:

1, this sentence is unclear Evaluation of the diagnostic and prognostic value of METTL7A in pan-cancer revealed that METTL7A had highly accuracy in tumor prediction and significantly correlated with the prognosis of kidney renal clear cell carcinoma (KIRC). Are the high or low expression, or the mutation of METTL7A significantly correlated with the poor prognosis?

2,  but also differential expressed in 7 molecular cancer subtypes and 8 im-mune cancer subtypes” Could you explain why you choose these subtypes and character of molecular cancer subtypes?

3, For figure 1, 40 targeted binding proteins of METTL7A were showed. It is interesting to find the interaction proteins of METTL7A. Could you show the type of 40 protein such as transcription factor etc? Do you think how many proteins interact with METTL7A in the cell? Is there other proteins other than 40 interaction with it?

4,  Figure 2 showed the differential expression of METTL7. what is the mechanism behind the high or low expression in the different tumor?

5, The resolution of figure 3 is low and not clear. So please provide higher resolution figures.

6, the primary therapy outcome, PR, SD showed only 1 or 2 sample. Is this too small number for P value accumulation?

7, The table 1, should the n(%) in the middle?

8, in the discussion part, the authors showed most of sentences that are repeat of results for the analysis. In my opinion, there should be more discussion about mechanism behind these results. So it is better to in-depth discuss, not just repeat the results part.

9, there are many association analysis between METTL7A and different cancer, however, in some extent, the relationship, similarities and difference are not clear. Perhaps, authors could revise in more logic way in the manuscript.

Reviewer 2 Report

The authors of "Evidence based on an integrative analysis of multi-omics data on METTL7A as a molecular marker in pan-cancer" systemically assessed the METTL7A’s function by integrating multi-omics data. Overall, this manuscript is well written and easy to follow, one of the major shortcomings is that there is no experimental validation for the potential function of METTL7A where the manuscript proposed. Thus, this work is a pure in silico exploration using public data and databases.

My comments are the following.

1. What is the key novelty of this work? In other words, what is the major contribution of this study to the field? It seems that once authors changed to another randomly selected gene instead of METTL7A, they can do the same analysis, and they will have more or less interesting findings.

2. In section 2.2, can the authors provide the number of samples in the TCGA and GTEx projects, respectively? As far as I know, there are ~10,000 samples in TCGA RNA-Seq data and ~10,000 or even more samples in GTEx RNA-Seq data. Please double-check that. 

3. I cannot find the survival data sources in the manuscript, for OS, DSS, and PFI.

4. In section 2.5, why the authors must convert PFKM value to TPM value?

5. In section 3.2 expression analysis, what the readers will learn to know METTL7A is upregulated in 8 cancer types and downregulated in 18 cancer types? What is the biological interpretation of the finding here?

6. In figure 2C. Is it necessary to analyze all the matched TCGA cancer samples? I think you can set a minimum cutoff to rule out cancer types with few matched samples. This point also may apply to figure 2D. Besides the results of the analysis, the authors are supposed to show the biological mechanism behind it.  

7. In section 3.8, did the authors perform an enrichment analysis on the merged DEGs? It is supposed to perform enrichment analysis on the upregulated/downregulated DEGs, respectively.

It is not reasonable to do enrichment analysis on the merged datasets.

8. The major problem of is manuscript is the lack of key findings. The authors did a lot of analyses and have tons of biological observations based on bioinformatics tools and databases. After carefully reading this manuscript, I even did not get any take-home message. It will be beneficial if the authors can focus on several important ones. I think the authors need to emphasize bullet findings in more detail so that the significance of the study is appreciated by a broader audience.

Round 2

Reviewer 1 Report

The revised manuscript solved my questions, and I think that it should be accepted. 

Reviewer 2 Report

The authors have made their best efforts to address my comments from the previous round of review. I have no more suggestions.